# Hydroxyl-Group Identification Using O K-Edge XAFS in Porous Glass Fabricated by Hydrothermal Reaction and Low-Temperature Foaming

**DOI:** 10.3390/molecules24193488

**Published:** 2019-09-26

**Authors:** Masanori Suzuki, Shigehiro Maruyama, Norimasa Umesaki, Toshihiro Tanaka

**Affiliations:** Graduate School of Engineering, Osaka University, 2-1 Yamadaoka, Suita, Osaka 565-0871, Japan; shigehiro.maruyama@mat.eng.osaka-u.ac.jp (S.M.); umesaki@mat.eng.osaka-u.ac.jp (N.U.); tanaka@mat.eng.osaka-u.ac.jp (T.T.)

**Keywords:** soft X-ray absorption, hydroxyl group, hydrothermal reaction, borosilicate glass, porous structure

## Abstract

Porous glass was prepared by the hydrothermal reaction of sodium borosilicate glass, and oxygen-ion characterization was used to identify the hydroxyl groups in its surface area. A substantial amount of “water” was introduced into the ionic structure as either OH^−^ groups or H_2_O molecules through the hydrothermal reaction. When the hydrothermally treated glass was reheated at normal pressures, a porous structure was formed due to the low-temperature foaming resulting from the evaporation of H_2_O molecules and softening of the glass. Although it was expected that the OH^−^ groups would remain in the porous glass, their distribution required clarification. Oxygen K-edge X-ray absorption fine structure (XAFS) spectroscopy enables the bonding states of oxygen ions in the surface area and interior to be characterized using the electron yield (EY) and fluorescence yield (FY) mode, respectively. The presence of OH^−^ groups was detected in the O K-edge XAFS spectrum of the porous glass prepared by hydrothermal reaction with a corresponding pre-edge peak energy of 533.1 eV. In addition, comparison of the XAFS spectra obtained in the EY and FY modes revealed that the OH^−^ groups were mainly distributed in the surface area (depths of several tens of nanometers).

## 1. Introduction

Metallurgical processes produce a large amount of slag and waste glass as by-products, which consist of various oxide components such as SiO_2_, CaO, Al_2_O_3_, and Na_2_O. Although these materials are currently recycled in road or concrete materials, additional value should be added to them for further beneficial use. 

“Exergy” determines the value of a material to be used (*Exergy* = $\Delta H-T_{0}\cdot \Delta S$, ΔH: enthalpy of the material, ΔS: entropy of the material, $T_{0}$: room temperature). It is easily assumed that waste slag and glass possess quite low exergies because the enthalpy of oxides is much lower than that of reduced metals and the entropy of a mixture of components is significantly higher than that of purified materials. However, the authors have proposed that the introduction of “interfaces” enhances the exergy of even waste slag and glass [1,2], which means that these reagents with low exergies could be transformed into value-added porous ceramic materials such as heat insulators, water-retentive materials, or filters for liquid or gas purification. 

Among various processes used to produce ceramic materials, hydrothermal reaction is a promising approach to fabricate porous glass materials as it can be operated with low energy consumption and the use of only water as an eco-friendly reactant [1,2,3,4,5,6,7]. Matamoros-Veloza et al. [3,4] previously proposed porous glass fabrication by the hydrothermal reaction as follows: first, highly pressurized water or steam at 473–573 K is used to dissolve sodium silicate glass components, which results in H_2_O-bearing glass production. Second, reheating of the H_2_O-bearing glass at normal pressures spontaneously creates a porous structure in the glass via the foaming phenomenon, which occurs after the vaporization of H_2_O incorporated in the glass and the softening of the glass structure. In general, softening of the glass occurs above the glass-transition temperature. Nakamoto and one of the authors [5,6] observed that the glass-transition temperature of sodium silicate glass significantly decreases as the apparent H_2_O content in the glass increases after hydrothermal reaction. They also reported that the B_2_O_3_-added glass exhibited the highest apparent H_2_O solubility and thus the lowest glass-transition temperature of all the glasses ever examined. Then, Yoshikawa and one of the authors showed that porous glass was successfully obtained by the hydrothermal reaction of the sodium borosilicate glass at 473 K and reheating of the H_2_O-bearing glass above 473 K because of its low-temperature foaming behavior [7,8]. To date, the authors have proposed several applications of the above porous glass as a carrier for ultrafine metal particles [9] or hydrate crystals [10,11]. Besides, there have been many works that applied hydrothermal treatment for glass recycling to fabricate advanced materials [12,13,14] and surface structure modification of glass materials [15,16,17].

Because the porosity and pore distribution of the above porous glass significantly depend on the apparent H_2_O content in the hydrothermally treated glass, it is important to clarify the mechanism of H_2_O incorporation in the ionic structure of sodium borosilicate glass during the hydrothermal reaction. Generally, the ionic structure of silicate glass consists of the linkage of SiO_4_ tetrahedra connected with each other through bridging oxygen ions (O^0^) and partially broken by non-bridging oxygen ions (O^−^) bonded with basic cations such as Na^+^ [18,19]. It is known that B_2_O_3_ contributes to enhance the linkage in the silicate glass ionic structure as BO_3_ triangles or BO_4_ tetrahedra [20]. In terms of H_2_O incorporation into the glass, however, it has only been reported for sodium silicate glass that H_2_O exists in the ionic structure either as OH^−^ groups coordinated with SiO_4_ tetrahedra (e.g., Si–OH) or isolated H_2_O molecules [21]. Although a substantial amount of H_2_O is introduced into the sodium borosilicate glass under the hydrothermal reaction, the population of the OH^−^ groups and H_2_O molecules and their distribution in the ionic structure of the glass have not yet been clarified. The OH^−^ groups in the glass ionic structure promote the softening of the glass as they disconnect the linkage of SiO_4_ or BO_4_ tetrahedra, whereas the interstitial H_2_O molecules are vaporized to form the porous structure when the H_2_O-bearing glass is reheated. In addition, neither the existence of residual OH^−^ groups nor its dispersion in the porous glass microstructure has been clarified.

Mid-infrared (mid-IR) absorption spectroscopy is normally used for identification of the hydroxyl (OH^−^) groups bonded with SiO_4_ tetrahedra and water molecules (H_2_O) in ceramic and glass materials [22,23]. However, it is difficult to distinguish them in the mid-IR spectrum because of the small difference between the corresponding absorption bands. In contrast, it has been confirmed that near-IR absorptions associated with the OH^−^ group or with the isolated H_2_O molecule are widely separated with each other and thus clearly recognizable [24], although reported trials of near-IR absorption spectroscopy of oxide materials are limited because of its low sensitivity.

The aim of the present study was to identify the hydroxyl group and its in-depth distribution in the porous glass made by the foaming of the H_2_O-bearing glass prepared by the hydrothermal reaction of the sodium borosilicate glass using both near-IR and oxygen K-edge soft X-ray absorption near-edge structure (XANES) spectroscopies. O K-edge XANES as well as O 1s X-ray photoelectron spectroscopy (XPS) have been previously used to identify the bonding state of oxygen ions with surrounding ions in oxide glass and crystals [25,26,27,28,29]. The O K-edge XANES spectrum of an oxide material provides not only the classification of O^0^ and O^−^ ions but detailed information on the bonding and coordination states of oxygen ions using the absorbed X-ray energy for electron excitation, while O 1s XPS does only former of the above using the corresponding binding energies. In addition, O K-edge XANES has been used for hydroxyl group speciation adsorbed on the surface of minerals, metals, and nanocrystals [30,31,32,33,34]. The hydroxyl group involved in the silicate network structure (Si–OH) can be classified as a non-bridging oxygen partially associated with the hydrogen cation, which could be recognized as an independent X-ray absorption in the O K-edge XANES spectrum with high resolution. Furthermore, the soft X-ray absorption measurements in both electron yield (EY) and florescence yield (FY) modes enable structural information in the surface and bulk areas to be separately obtained. 

## 2. Results

### 2.1. Preparation of H_2_O-Bearing Glass by Hydrothermal Reaction and Fabrication of Porous Glass by Low-Temperature Foaming

Figure 1a shows the appearance of the H_2_O-bearing glass prepared by the hydrothermal reaction of the sodium borosilicate glass (see Section 4 for chemical composition of the original glass and detailed hydrothermal conditions). The sodium borosilicate glass components reacted with the highly pressurized H_2_O vapor (the partial pressure of H_2_O was approximately 1.6 MPa at 473 K according to the H_2_O phase diagram) to form a homogeneous aqueous solution. After drying the sample to remove excess condensed water, the solidified transparent H_2_O-bearing glass species was obtained. Approximately 30 mass % of weight increase was observed after the hydrothermal treatment, which was partially due to the H_2_O incorporation into the glass.

Figure 1b shows the appearance of the porous glass obtained by reheating the H_2_O-bearing glass in a microwave oven. When reheated, the H_2_O-bearing glass largely expanded to form a porous structure because of the foaming behavior caused by both the softening of the H_2_O-bearing glass and the vaporization of H_2_O inside the glass (see Appendix A as an evidence of interstitial H_2_O emission from the hydrothermally treated glass).

Figure 1c shows the cross-sectional microstructure of the porous glass. A three-dimensional broken-bubble structure is observed, with the size of the bubbles varying from ten to several hundred of micrometers in diameter. The shells of the bubbles are made of the sodium borosilicate glass containing H_2_O, and the walls of broken bubbles are partially interconnected to form the porous structure.

### 2.2. Near-IR Absorption Spectroscopy for Identification of OH^−^ Groups and H_2_O Molecules in Porous Glass

The near-IR absorption of the original sodium borosilicate glass, H_2_O-bearing glass, and porous glass was examined using diffuse reflectance spectra. Figure 2 presents the near-IR absorbance spectra as a function of wavenumber of incoming light. The Kubelka–Munk function *F* denotes the ratio of the absorption and reflection degrees:
(1)$$F=\frac{{\left(1-R\right)}^{2}}{2R}$$
where *R* represents the absolute spectral reflectivity of the material. An increase of *F* directly corresponds to an increase of absorbance.

It has been mentioned that the absorptions in the wavenumbers between 6400 and 7400 cm^−1^ are associated with the first overtone of the hydroxyl stretching modes (OH^−^ groups), whereas those between 4800 and 5400 cm^−1^ correspond to water combination modes (H_2_O molecules) [24]. The near-IR spectrum for the original sodium borosilicate glass did not indicate the presence of either OH^−^ groups or H_2_O molecules. In contrast, the spectrum for the H_2_O-bearing glass clearly exhibited both of the IR absorptions associated with hydroxyl groups and H_2_O molecules. For the porous glass obtained by reheating the H_2_O-bearing glass, decreased absorptions associated with the OH^−^ group were observed in the spectrum, whereas the IR absorption corresponding to the H_2_O molecules present was too small to be clearly recognized. Thus, it was verified that the H_2_O-bearing glass contained many OH^−^ groups and H_2_O molecules in its ionic structure, and that most of the H_2_O molecules and partial OH^−^ groups were removed from the porous glass after reheating.

### 2.3. O K-Edge XANES Spectroscopy of Porous Glass to Determine in-Depth Distribution of OH^−^ Groups

Figure 3 represents the O K-edge XANES spectra of the original sodium borosilicate glass and the porous glass prepared using the hydrothermal reaction and foaming behavior as well as those for SiO_2_ and SiO_2_–Na_2_O glasses as reference materials. The partial electron yield (PEY) mode was selected for collection. The spectrum for SiO_2_ glass, which contains only bridging oxygen ions (O^0^), consisted of a single absorption peak at 537.5 eV (marked as E_3_ in Figure 3). In contrast, the spectrum for SiO_2_–Na_2_O glass, where both bridging oxygen (O^0^) and non-bridging oxygen ions (O^−^) are included, consisted of both the pre-edge absorption peak at 531.6 eV (marked as E_1_) and the main absorption peak E_3_. Therefore, it is verified that the absorbed peak energy E_1_ corresponds to the non-bridging oxygen ions (Si–O–Na), whereas E_3_ corresponds to the bridging oxygen ions (Si–O–Si). The spectrum for the original sodium borosilicate glass also contained both the E_1_ and E_3_ absorption peaks, indicating that the glass contains both O^0^ and O^−^ ions in its ionic structure. In contrast, a unique absorption peak (marked as E_2_) was observed in the spectrum for the porous glass as well as the E_1_ and E_3_ absorption peaks. The peak energy for E_2_ was identified as 533.1 eV, which corresponds to the OH^−^ group, as reported by several authors [31,32,33,34]. 

Figure 4 presents the O K-edge XANES spectra of the porous glass obtained using different collection modes. The partial electron yield (PEY) and total electron yield (TEY) modes reflect the ionic structure in the surface area (depths of 50 nm or less from the surface), whereas the total florescence yield (TFY) mode reflects that in the bulk area (depths of greater than 50 nm from the surface). The PEY mode detects the ionic structure at very shallow depths from the surface (depths of 10 nm or less) because of the screening effect, whereby only the radiated Auger electrons with high kinetic energies are detected. In Figure 4, we can assume that the peak energies E_1_, E_2_, and E_3_ are associated with non-bridging oxygen (e.g., Si–O–Na), hydroxyl groups (e.g., Si–OH), and bridging oxygen (e.g., Si–O–Si), respectively, each coordinated with a silicon cation, from the results in Figure 3. The spectra obtained in PEY and TEY modes clearly indicate the existence of the hydroxyl groups. In particular, the corresponding absorption peak in the TEY spectrum was remarkable. In contrast, the absorption peak E_2_ corresponding to the hydroxyl groups was not notable in the TFY spectrum.

## 3. Discussion

Concerning the OH^−^ group identification in the porous glass by O K-edge XANES spectroscopy (please see Figure 3), we focused on the relationship between the peak energy and the bond strength between a cation (M^n+^) and oxygen anion, where we assumed that each oxygen makes the other bond with a silicon cation. If we assume that the absorbed X-ray energy, which reflects electronic state of valence electrons in oxygen elements associated with cations, is proportional to the energy for the M–O bond dissociation, the peak energy E_2_ corresponds to the hydroxyl group associated with the silicon cation (Si–OH) because the bond dissociation energies (*D*) of the M–O diatomic molecule defined by Luo [35] satisfy the following relationship: *D*(Na–O) < *D*(H–O) < *D*(Si–O), corresponding to the peak energy relationship E_1_ < E_2_ < E_3_. Thus, the use of O K-edge XANES spectroscopy enabled the characterization of oxygen ions bonded with different ions, particularly the OH^−^ group speciation in the porous glass created by the hydrothermal reaction of sodium borosilicate glass. 

In addition, the O K-edge XANES measurements performed using different collection modes enabled the evaluation of the in-depth distribution of the OH^−^ groups in the porous glass microstructure (see Figure 4). Here, it should be noted that although all the samples were prepared as powders, we can assume that the spectra obtained in PEY and TEY modes mainly reflect the ionic structure of the pore surface rather than the cross-sectional planes because of the adequately large pore surface area. The results in Figure 4 indicate that the hydroxyl groups were selectively distributed in the pore surface (up to 50 nm in depth) rather than in the interior of the porous glass microstructure. This finding is of importance in our understanding of how the porous microstructure is formed after the vaporization of H_2_O molecules in the ionic structure of the glass when the foaming behavior occurs. When the H_2_O-bearing glass is heated at normal pressure, the incorporated H_2_O molecules in the ionic structure are immediately saturated, and then, the emission of H_2_O vapor most likely occurs by moving out of the linkage of SiO_4_ or BO_4_ tetrahedra disconnected by OH^−^ groups to form a bubble (Figure 5 shows a schematic image of H_2_O behavior in glass microscopic structure), in addition, softening of the glass structure occurs. This process may result in the segregation of hydroxyl groups in the surface area of the broken-bubble structure made of the softened glass. Thus, for preparation of a largely expanded porous glass, the optimum glass composition should be determined to enhance the disconnection of the silicate tetrahedra linkage by hydroxyl group incorporation as well as H_2_O molecules entrapment during the hydrothermal reaction. 

## 4. Materials and Methods

Sodium borosilicate glass with the chemical composition 63 SiO_2_–27 Na_2_O–10 mass% B_2_O_3_ was synthesized by mixing powder reagents of silicon dioxide, sodium carbonate (special grade of FUJIFILM Wako Pure Chemical Corporation, Osaka, Japan), and boron trioxide (special grade of Mitsuwa Chemicals Co. Ltd., Osaka, Japan) in a mortar, melting the mixture in a Pt–Rh crucible at 1673 K in air, and pouring the melt onto a Cu plate for quenching. This glass composition was selected on the basis of the report by Nakamoto et al. [5] to maximize the apparent H_2_O content by the hydrothermal reaction. The sodium borosilicate glass was subjected to hydrothermal reaction by putting 1 g of the glass sample in the powder state and 5 mL of purified water in a sealed autoclave and heating the autoclave at 473 K for 2 h. The glass sample was placed in an inner Teflon crucible (inner diameter: 17 mm, height: 27 mm) and separated with water placed in an outer Teflon container (inner diameter: 30 mm, height: 40 mm) in the autoclave. The detailed conditions of the autoclave setup for the hydrothermal reaction can be found elsewhere [10]. After the hydrothermal reaction, the resulting sample was dried in the oven at 353 K overnight to remove the condensed water, and then, the H_2_O-bearing glass was obtained. The H_2_O-bearing glass was then reheated in a 1000-W microwave oven (Model ER-J6, Toshiba Corporation, Tokyo, Japan) for 30 s to make a porous glass after the foaming behavior. The X-ray diffraction analysis was performed for powders of the original glass, the hydrothermally treated glass and the porous glass (see Appendix A), and it was confirmed that all of the samples were glassy and no crystalline phases were found.

The near-IR absorbance of each sample was evaluated by measuring the relative diffuse reflectance using a UV–Vis–NIR spectrophotometer (Model UV-3600, Shimadzu Corporation, Kyoto, Japan) in the wavelength range between 730 and 2500 nm; barium sulfate powder was used as the reference material. The absolute spectral reflectivity of each sample was then determined by multiplying the absolute spectral reflectivity of the reference material by the measured relative reflectance, where the absolute reflectivity data of barium sulfate powder was obtained from the report by Grum et al. [36]. 

Oxygen K-edge XANES measurement for each sample was conducted at beamline BL-11 at the Ritsumeikan University SR Center, Shiga, Japan. Each sample was prepared as a powder, which was placed on an indium sheet attached to a holder with conductive carbon tape. The holder with the sample was then placed in the transfer vessel. The above preparation was performed in a glove box placed under vacuum and then filled with purified Ar gas to reduce moisture in the atmosphere. After the vessel was set, the X-ray absorption spectroscopy measurement apparatus originally built in the SR center was placed under vacuum at 10^−6^ Pa. Finally, the O K-edge XANES spectra were measured in the range between 520 and 600 eV using a KTP (011) spectroscopic crystal, where the resolution of the X-ray energy (ΔE/E) was approximately $6.5\times 10^{-4}$.

## Figures and Tables

**Figure 1 molecules-24-03488-f001:**
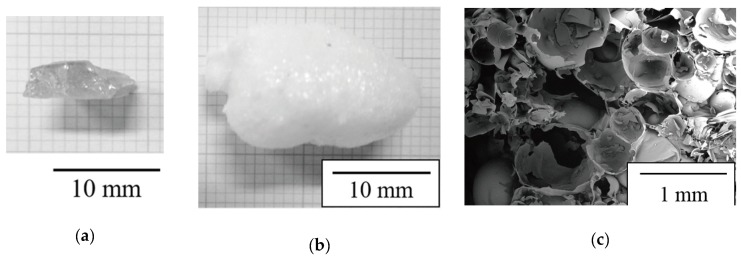
(**a**) Appearance of H_2_O-bearing glass after hydrothermal reaction. (**b**) Appearance of porous glass obtained by reheating the H_2_O-bearing glass. (**c**) Cross-sectional SEM image of microstructure of porous glass.

**Figure 2 molecules-24-03488-f002:**
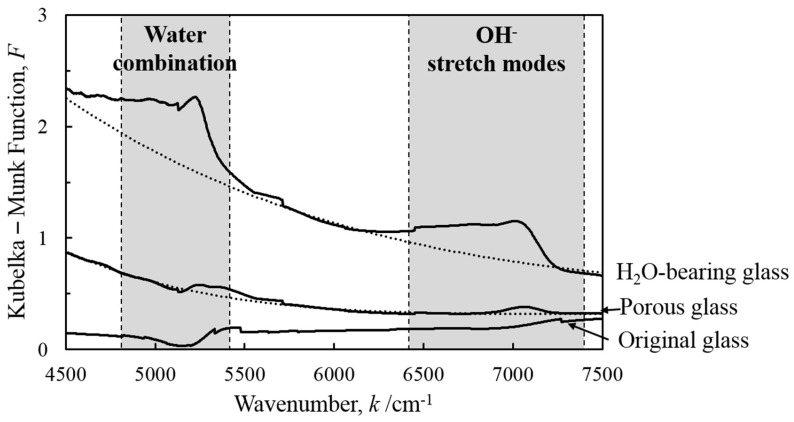
Near-IR absorbance spectra of glass samples. The solid lines are the observed spectra, and the dashed lines are fitted baselines estimated using polynomial functions.

**Figure 3 molecules-24-03488-f003:**
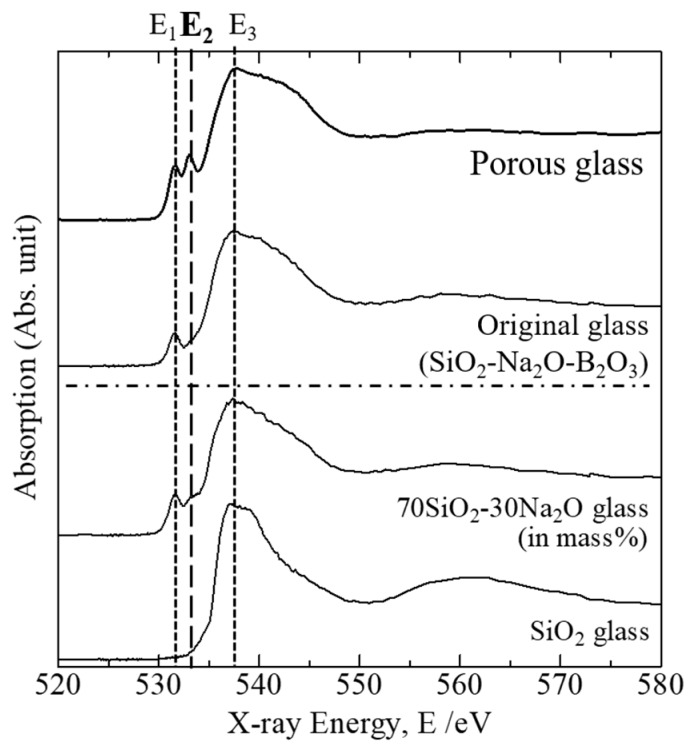
Oxygen K-edge X-ray absorption near-edge structure (XANES) spectra of porous glass and original sodium borosilicate glass observed in partial electron yield (PEY) mode. The spectra for SiO_2_ and 70% SiO_2_–30 Na_2_O (in mass %) glasses are provided as reference materials for non-bridging oxygen and bridging oxygen ions.

**Figure 4 molecules-24-03488-f004:**
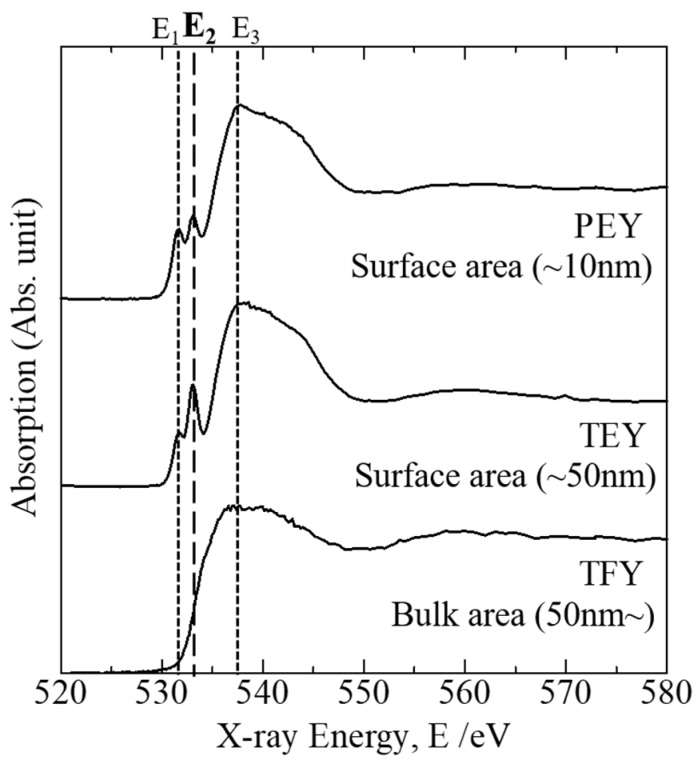
Oxygen K-edge XANES spectra of porous glass obtained using different collection modes. The PEY and TEY modes reflect the ionic structure in surface area, whereas the TFY mode reflects that in the bulk.

**Figure 5 molecules-24-03488-f005:**
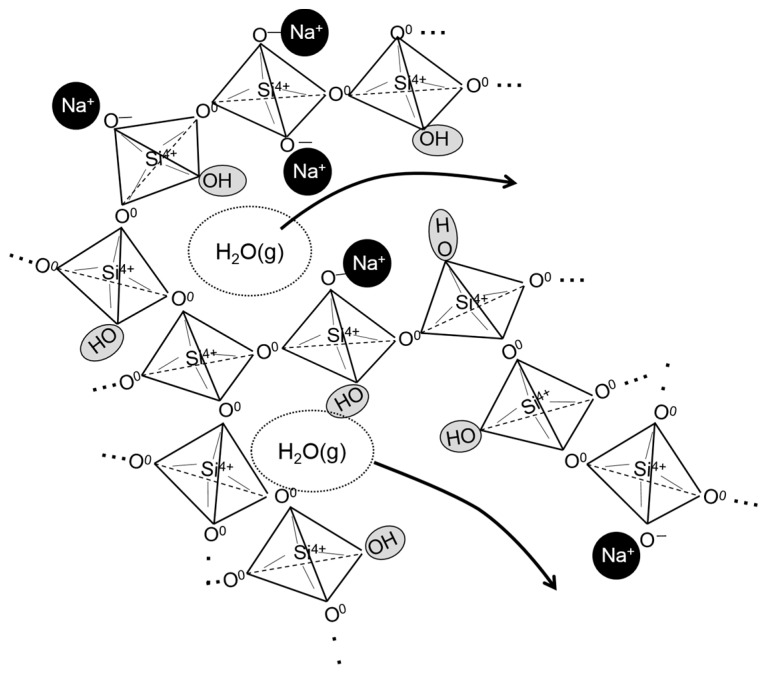
Schematic illustration of H_2_O vapor emission from glass microscopic structure when reheating the hydrothermally treated glass.

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
