# Peer review of "Hydroxyl-Group Identification Using O K-Edge XAFS in Porous Glass Fabricated by Hydrothermal Reaction and Low-Temperature Foaming"

_molecules, 2019, doi:10.3390/molecules24193488_

Round 1

Reviewer 1 Report

In this manuscript, Masanori Suzuki et al tried to identify the presence of H2O and -OH within the hydrothermally treated glass using O K-edge XAFS technique. In general, the manuscript is well written and the results are well presented. However, the authors should revise their manuscript based on the following points, before it can be considered for publication.

The citation of literatures in the manuscript should be improved. In the second paragraph, no literature is cited when the authors review the possible applications of the hydrothermally recycled glass. Some of the literatures cited in the introduction section are too old, considering there are many new works published recently. The authors should also give proper credit to similar works that are published more recently. Some of the more recent important literatures should be cited, for example: ChemistrySelect 2018, 3, 11494; Small, 2017, 13, 1700391. Journal of American Ceramic Society, 99, 128. Journal of Non-Crystalline Solids, 2016, 452, 93. In line 104-105, the authors claim the water content in the hydrothermally treated glass is 30% based on the weight increase. It is not accurate to estimate the water content based on the weight increase, because the hydrothermal treatment will etch the silica, resulting in the leaching of SiO2 into the water. The authors should re-characterize the water content using TGA method. From Figure 1c, it is hard to identify the “broken-bubble structure”. I think the SEM image indicates the formation of new species in a flake-like shape. If the authors want to claim such structure is caused by the vaporization of the interstitial water, more evidence has to be provided. It is very important that the author must provide XRD data of the glass before and after the hydrothermal treatment, in order to prove that no other new compound has been formed during the hydrothermal treatment. It is not necessary to include the supplementary information (SI), which only consist one figure. The figure is not the data obtained by the author, and the phase diagram of water is a common knowledge.

Author Response

Dear Reviewer,

we thank to your precious comments and kind suggestions to our submitted manuscript. The response to each of your comment is presented below. In the revised manuscript, the modified parts are underlined and yellow-highlighted.

1) The citation of literatures in the manuscript should be improved. In the second paragraph, no literature is cited when the authors review the possible applications of the hydrothermally recycled glass. Some of the literatures cited in the introduction section are too old, considering there are many new works published recently. The authors should also give proper credit to similar works that are published more recently. Some of the more recent important literatures should be cited.

> The second paragraph in Introduction represented our concept for the recycling of waste slag and glass to advanced porous materials on the basis of increasing exergy of a material, where the means have not been limited to hydrothermal treatment. In the third paragraph we introduced hydrothermal treatment as a beneficial method to fabricate porous materials, so we added a statement there to introduce several recent works concerning glass recycling and its surface structure modification by hydrothermal treatment, as follows:

In the revised manuscript, line 56-57:

Besides, there have been many works that applied hydrothermal treatment for glass recycling to fabricate advanced materials [12-14], and surface structure modification of glass materials [15-17].

The corresponding references are listed below:

12. Elmes, V.K; Edgar, B.N; Mendham, A.P; Coleman, N.J. Basic metallosilicate catalysts from waste green container Ceram. Int. 2018, 44, 17069-73.

13. Gattullo, C.E; D’Alessandro, C; Allegretta, I; Porfido, C; Spagnuolo, M; Terzano, R. Alkaline hydrothermal stabilization of Cr(VI) in soil using glass and aluminum from recycled municipal solid wastes. Hazard. Mater. 2018, 344, 381-9.

14. Kamitani, M; Tagami, T; Fukuya, T; Kondo, M; Hiki, T; Nakahira, A. Synthesis of A- Type zeolite from flat glass recycle by hydrothermal treatments and its evaluation. Key Eng. Sci. 2014, 616, 183-7.

15. Ma, Q; Cheng, H; Yu, Y; Huang, Y; Lu, Qipeng; Han, Shikui; Chen, J; Wang, R; Fane, A.G; Zhang, H. Preparation of Superhydrophilic and Underwater Superoleophobic Nanofiber-Based Meshes from Waste Glass for Multifunctional Oil/Water Separation. Small 2017, 13, 1700391.

16. Duraisamy, S; Priyadarshini, B.G. Enhancing the Optical Behavior of Glass Surface by Creation of Microstructures in Single‐Step Hydrothermal Wet Etching. ChemistrySelect 2018, 3, 11494-504.

17. Luo, J; Huynh, H; Pantano, C.G; Kim, S.H. Hydrothermal reactions of soda lime silica glass – Revealing subsurface damage and alteration of mechanical properties and chemical structure of glass surfaces. J. Non-Cryst. Solids 2016, 452, 93-101.

2) the authors claim the water content in the hydrothermally treated glass is 30% based on the weight increase. It is not accurate to estimate the water content based on the weight increase, because the hydrothermal treatment will etch the silica, resulting in the leaching of SiO2 into the water. The authors should re-characterize the water content using TGA method.

> We previously performed TG-DTA analysis to more precisely evaluate H2O content in the hydrothermally treated glass, and we confirmed that the estimated H2O content from weight change after hydrothermal treatment was comparable to that measured by the TG-DTA analysis. Therefore, we considered that the effect of silica etching during hydrothermal reaction on the glass sample weight loss was small. However, for the current hydrothermal condition we did not take the TG-DTA analysis, so the apparent H2O content was just estimated from the weight change. In the revised manuscript, the corresponding sentence was modified to the following:

In the revised manuscript, line 105-106:

Approximately 30 mass% of weight increase was observed after the hydrothermal treatment, which was partially due to the H2O incorporation into the glass.

3) From Figure 1c, it is hard to identify the “broken-bubble structure”. I think the SEM image indicates the formation of new species in a flake-like shape. If the authors want to claim such structure is caused by the vaporization of the interstitial water, more evidence has to be provided.

> We replaced Figure 1c to another figure where the broken-bubble structure can be more clearly found. A flake-like shape species in the previous figure were recognized as a fragment of the broken-bubble structure because the shape was very random and dispersed inhomogeneously. 

A new Figure S1 was inserted to supplementary materials as an additional evidence to prove the emission of interstitial water, where the porous glass was expanded in a transparent glass tube and the condensed water droplets were attached on the inner wall of the tube. In the revised manuscript, the following sentence was inserted:

In the revised manuscript, line 110-111:

(see Figure S1 in supplementary materials as an evidence of interstitial H2O emission from the hydrothermally treated glass).

In the revised manuscript, line 241-242:

Supplementary Materials: Figure S1: Porous glass expanded in a transparent glass tube, showing condensed water emitted from the hydrothermally treated glass when heated.

4) It is very important that the author must provide XRD data of the glass before and after the hydrothermal treatment, in order to prove that no other new compound has been formed during the hydrothermal treatment.

> The result of X-ray diffraction analysis for the glass materials has been inserted as Figure S2 in supplementary materials, where the samples are verified to be glassy and no crystalline phase are not detected after hydrothermal treatment or reheating. However, the shape of hallo patterns are found to be changed after the hydrothermal treatment, which may indicate some degrees of microscopic structural change in the glass.

In the revised manuscript, the following sentence was inserted:

In the revised manuscript, line 223-225:

The X-ray diffraction analysis was performed for powders of the original glass, the hydrothermally treated glass and the porous glass (see Figure S2 in supplementary materials), and it was confirmed that all of the samples were glassy and no crystalline phases were found.

In the revised manuscript, line 242-244:

Supplementary Materials: Figure S2: X-ray diffraction patterns of sodium borosilicate glass powders as prepared, after hydrothermal treatment and reheating by microwave oven (porous glass), where Cu Kα radiation was used.

5) It is not necessary to include the supplementary information (SI), which only consist one figure. The figure is not the data obtained by the author, and the phase diagram of water is a common knowledge. 

> The phase diagram of water has been removed from the supplementary materials, instead the above two figures have been included.

Reviewer 2 Report

The authors have reported the distribution of hydroxyl groups and water molecules within the porous glass formed by hydrothermal reaction. So far, the identification of hydroxyl units in glasses was done by mid-IR absorption spectroscopy, and the authors have used XAFS tools to further identify the hydroxyl group speciation in the porous glass. The manuscript is relatively well written but the language correction is required. This manuscript is recommended for publication in Molecules after text editing. 

Author Response

Dear reviewer,

We thank to your kind suggestion. In this paper, we performed near-IR absorption spectroscopy rather than mid-IR to more clearly distinguish hydroxyl group and H2O molecules incorporated in the porous glass from the corresponding IR absorptions. In addition, O K-edge XANES spectroscopy was performed not only to identify hydroxyl groups partially associated with silicate network structure but to obtain an evidence of the hydroxyl groups mainly distributed in the surface area from the comparison of the spectra by EY and FY modes.

The English of the original manuscript was carefully edited by an expert before submission, as stated in the last sentence in acknowledgements. However, English of the manuscript was carefully checked again and partially revised.

Round 2

Reviewer 1 Report

The authors have carefully addressed my comments, and the required experiments were performed and presented in the revised manuscript. I have no further questions or comments. This manuscript can be accepted in its current form.